# CLF: Curve Line Fitting Neural Network Based On Bezier Curve

## Abstract

The Multilayer Perceptron (MLP) serves as a fundamental architecture in deep learning, leveraging the universal function approximation theorem through linear regression combined with activation functions. Despite its widespread use, the inclusion of activation functions contributes to the inherent nature of MLPs as "black boxes," limiting their interpretability. In this paper, we propose a novel Curve Line Fitting (CLF) network, which introduces Bezier curve fitting to directly address nonlinear distributions. By replacing traditional linear regression with Bezier curve regression, the CLF network offers a more efficient means of fitting target distributions. Additionally, the removal of activation functions makes the CLF model fully interpretable, enabling clear insights into the relationships between input dimensions and target distributions, as well as the interdependencies across different dimensions. (Sample code for the CLF model will be made available on GitHub.)

## 1 Introduction

The MLP [Haykin (1998); Cybenko (1989); Hornik et al. (1989a)] is a widely used network structure in deep learning due to its ability to efficiently approximate any target distribution. It effectively employs the universal function approximator theorem by using linear regression and activation functions [He & Xu (2024); Hornik et al. (1989b)]. Consequently, many advanced network architectures incorporate MLP as a fundamental component [Targ et al. (2016); Vaswani et al. (2023); Devlin et al. (2019); Li et al. (2018); Zhao et al. (2018) ]. Despite its widespread application, the MLP architecture is often considered "black box," leading to three significant challenges. First, determining the most efficient MLP structure for a specific target distribution is challenging [Bergstra & Bengio (2012); Ngoc et al. (2021)]. Second, when an MLP fails to converge, it is difficult to diagnose the underlying issue or implement a solution to ensure convergence [Várkonyi-Kóczy et al. (2014)]. Lastly, although an MLP may achieve high accuracy, it does not readily reveal the relationships between the input space and the target distribution, limiting interpretability.

Considerable research has been devoted to demystifying the "black box" nature of the MLP. Some approaches focus on enhancing the MLP structure itself, such as updating the activation functions to be learnable [Liu et al. (2024)], while others aim to decipher the specific knowledge that MLP acquires at each layer [Gorokhovatskyi et al. (2020)]. Although these efforts have yielded some progress in various aspects [He (2020), Xiang et al. (2005)], the three primary challenges still persist.

The MLP utilizes linear regression combined with activation functions to model complex relationships. While linear regression is straightforward and interpretable, the incorporation of activation functions introduces ambiguity into the network. To fundamentally address this limitation, this paper introduces the novel Curve Line Fitting (CLF) structure, which remove activation functions altogether, thereby enhancing the transparency and explainability of the network. With the removal of activation functions, the traditional linear regression approach proves inadequate for modeling complex distributions. We adopt Bezier Curve fitting as an alternative. Bezier Curves [Floater (1992)], defined by a set of control points, can approximate almost any shape, making them highly versatile for modeling diverse target distributions. Although multiple researchers have explored Bezier Curve fitting for single dimension target distributions [Shao & Zhou (1996), Mineur et al. (1998)], no existing network architecture has been based solely on this approach.

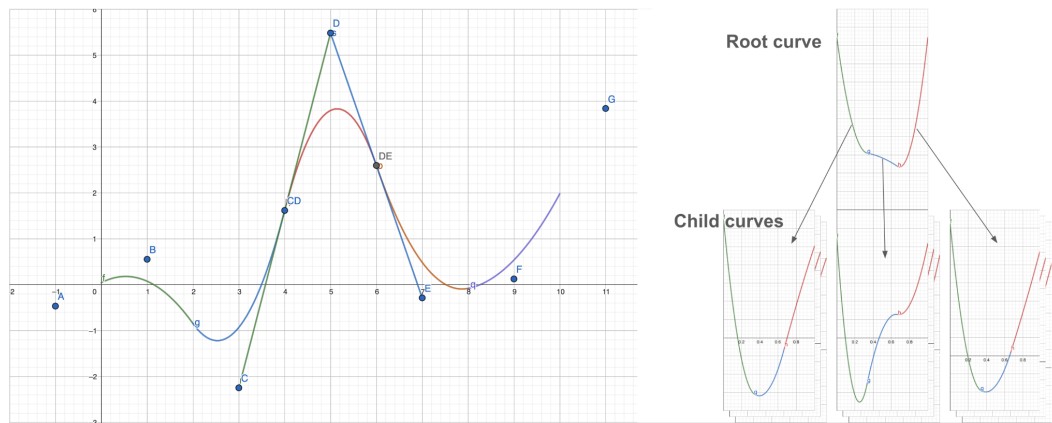

Figure 1: (a): Single-node CLF structure.          (b): Multi-layer CLF structure.

This paper introduces the CLF model, a novel approach that utilizes Bezier Curves to develop a multi-layer network structure. The CLF model offers two significant advantages: (1) It is fully explainable and capable of clearly demonstrating the relationships it learns. Upon completion of training, the CLF model can vividly illustrate both the relationship between the input space and target distribution, and the interactions among different input dimensions. (2) The explainability of the CLF model provides a clear guide during and after training. This transparency allows for an assessment of whether using fewer parameters could achieve comparable results, by analyzing the network's structure and performance. Additionally, there is only one known issue that can prevent the CLF from converging during training, which can be readily addressed by adjusting the CLF settings.

## 2 METHOD

CLF employs the Bezier Curve to fit the target distribution, primarily leveraging its capability to approximate any shape effectively [Floater (1992)]. Because Bezier Curves inherently fit nonlinear distributions, the activation function is not used in CLF. This section elaborates on how CLF adapts the target distribution across various configurations, including single-node, single-layer, and multi-layer architectures.

### 2.1 SINGLE-NODE CLF

Figure 1 (a) illustrates the representation of a nonlinear distribution by a single-node CLF. In this figure, $X \in [0, 10]$ represents the input space, and $Y$ denotes the target distribution. (Further details on the limitations related to the input space $X$ is discussed in the Appendix.) Specifically, input space $X$ is evenly divided into five segments, governed by control points labeled A through G. These control points are positioned with a learnable y-coordinate and a fixed x-coordinate ranging from $-1$ to $11$.

Focusing on segment curve $[4, 6]$, this curve is influenced by control points $C$, $D$, and $E$. More precisely, it is regulated by points $CD$, $D$, and $DE$, where $CD$ and $DE$ represent the initial and terminal points. The $CD$ and $DE$ are calculated as $CD = \frac{C+D}{2}$ and $DE = \frac{D+E}{2}$ respectively. According to the properties of Bezier Curves, the curve $[4, 6]$ at point $CD$ possesses the same derivative value as the straight line $[CD, D]$. Similarly, the curve $[2, 4]$ at point $CD$ maintains the same derivative value as the straight line $[C, CD]$. Therefore, the continuity and differentiability of the curve $[2, 6]$ at $x = 4(CD_x)$ are ensured. Consequently, the entire curve over the interval $[0, 10]$ is continuous and differentiable.

### 2.1.1 GET $\hat{\mathbf{y}}$

Previous example present that each segment curve is controlled by three control points, denoted as $P_1$, $P_2$, and $P_3$. The midpoints between $P_1$ and $P_2$ are calculated as $P_{12} = \frac{P_1+P_2}{2}$. Denote $s = \frac{input\ \ range}{segmentation\ \ number}$. Due to space constraints, the detailed derivation of the formulas is provided in the Appendix. The parameter $t$ and $\hat{\mathbf{y}}$ are:

$$t = \frac{x - P_{12x}}{s}; \hat{\mathbf{y}} = (\frac{P_1}{2} - P_2 + \frac{P_3}{2})t^2 + (-P_1 + P_2)t + \frac{P_1 + P_2}{2} \tag{1}$$

### 2.1.2 OPTIMIZATION FUNCTION

Derivative of equation (1) with respect to $P_1, P_2, P_3$, get $P_1' = \frac{1}{2}t^2 - t + \frac{1}{2}$, $P_2' = -t^2 + t + \frac{1}{2}$, and $P_3' = \frac{1}{2}t^2$, $loss = y - \hat{\mathbf{y}}$, learning rate (LR) is a hyper-parameter. Because the control points' x-positions are fixed, CLF only optimizes the control points' y-positions. The new $P_1, P_2, P_3$ y-positions are:

$$[P_1, P_2, P_3] = [P_1, P_2, P_3] + [P_1', P_2', P_3'] * loss * LR \tag{2}$$

Equation (2) shows that 1) Optimizing the control points only depends on $t$ and $loss$, which means CLF Optimization Function does not require backward function. 2) During Optimization Function, only a subset (2-3 parameters each dimension) of the network is optimized. Specifically, parameters closer to the current sample receive higher optimization values, whereas those further away are assigned lower or even zero optimization values. This optimization approach is analogous to neural processes in the brain, where only specific regions interact and respond to particular stimuli [Kolb & Whishaw (1998)].

### 2.1.3 TOQUADRATICLIST FUNCTION

The value of $\hat{\mathbf{y}}$ can be derived from Equation (1). However, each segment curve can alternatively be represented by a part of quadratic equation, which necessitates significantly fewer computational resources compared to Equation (1). The following outlines the process of transforming Equation (1) into its equivalent quadratic form.

Set $w_1 = \frac{P_1}{2} - P_2 + \frac{P_3}{2}$, $w_2 = -P_1 + P_2$, $w_3 = \frac{P_1+P_2}{2}$, $p = P_{12x}$

$$Equation(1) = \frac{w_1}{s^2}x^2 + (-\frac{2w_1 p}{s^2} + \frac{w_2 p}{s})x + (\frac{w_1 p^2}{s^2} - \frac{w_2 p}{s} + w_3) \tag{3}$$

Utilizing Equation (3), it is demonstrated that each segment curve, defined by three control points, can be transformed into a quadratic equation of the form: $\hat{\mathbf{y}} = ax^2 + bx + c$. This conversion allows for a simplified representation of the segment curves and facilitating easier computation.

### 2.1.4 FORWARD FUNCTION

Transforming the control points into a list of quadratic equations significantly enhances the forward function's computational efficiency. This function initially employs a mask, $x/s$, to determine the appropriate quadratic equation for a given input. Subsequently, it utilizes the selected quadratic equation to compute the $\hat{\mathbf{y}}$. This methodology streamlines the process, enabling faster and more efficient calculations within the network. This forward approach also mirrors cognitive processes in the human brain. When individuals tackle complex mathematical problems, they typically do not derive all relevant formulas from scratch; instead, they rely on memory to recall necessary formulas.

### 2.1.5 INITIALIZATION

To initialize a single-node CLF, we need to define the maximum value of the input space, $max$, and the number of segments, $seg$. The domain for the input space is set to $[0, max]$. The CLF model then generates a list of control points, represented as $conList = [A_y, B_y...] \in R^{seg+2}$, and a list of quadratic equations, represented as $equList = [[a, b, c]] \in R^{seg*3}$.

### 2.1.6 TRAINING

```
init: LR; [0, max]; seg; conList; equList
for x in X:
    ŷ = Forward(x) // use equList, ax² + bx + c, get ŷ
    loss = y − ŷ; Optimization(loss, LR)// update conList
    equList = ToQuadraticList(conList) // update equlist
```

## 2.2 SINGLE-LAYER CLF

### 2.2.1 SINGLE-OUTPUT

In an MLP, the output can be expressed as $w_1 x_1 + w_2 x_2 + ... + b$, where $w_1$, $w_2$ ... are weights computed from the MLP parameters and influenced by activation functions. Drawing inspiration from this framework, the single output CLF aggregates the results across all dimensions, yielding the output $\hat{\mathbf{y}} = \sum_{i=0}^n \hat{\mathbf{y}}_i = \sum_{i=0}^n f(x_i)$. This approach allows the CLF to integrate individual dimension contributions into a collective output, similar to the summation method used in MLPs.

### 2.2.2 MULTI-OUTPUT

The CLF is also inspired by the MLP for multi-output tasks such as taxonomy classification. The multi-output CLF utilizes multiple networks to compute each output independently, selecting the highest value index as the definitive result.

In the single-layer CLF, the control point list is modified to $conList \in R^{N*(seg+2)}$, and the quadratic equation list is modified to $equList \in R^{N*seg*3}$. This configuration effectively addresses the computation of outputs that are the sum of independent function variables, such as $y = f(x_1) + f(x_2)$. However, it is less effective for distributions that involve interactions between variables, such as $y = x_1 * x_2$. To overcome this limitation, a multi-layer CLF is proposed.

## 2.3 MULTI-LAYER CLF

Gradient boosting [Xiang et al. (2020)] is a machine learning technique wherein each iteration of the model seeks to fit the negative gradient of the residuals from the prior iteration, thereby systematically reducing the total loss with each subsequent round. Inspired by this principle, the CLF network architecture adapts and extends this concept within its multi-layer structure. Unlike gradient boosting focuses on fitting the negative gradient of residuals, the CLF involves different nodes fitting the negative loss of each other, facilitating a more pronounced reduction in the overall loss of the network.

The development of a multi-layer CLF entails three principal steps. Initially, a single-layer CLF is trained to establish a baseline understanding of the data. Subsequently, dimension relations are calculated using the data from the single-layer CLF, allowing for the grouping of related dimensions based on their interactions. Finally, a multi-layer CLF is constructed based on these dimension groups and then trained to model and predict complex interactions among the variables.

### 2.3.1 GROUP RELATED NODES

The training dataset $X$ has $M$ samples with $N$ dimensions, $X \in R^{M*N}$. The target data is represented as $Y \in R^M$. As discussed in the previous section, $\hat{\mathbf{y}}$ is the sum of individual predictions across all dimensions, $\hat{\mathbf{y}} = \sum_{i=0}^n \hat{\mathbf{y}}_i$. In this section, $\hat{\mathbf{Y}}_{all}$ refers to the array of predictions before summation, $\hat{\mathbf{Y}}_{all} \in R^{M*N}$. $\hat{\mathbf{y}}_{ij}$ represents the predicted value for the $i^{th}$ sample in the $j^{th}$ dimension. $\hat{\mathbf{y}}_{:,i}$ indicates all predictions for the $i^{th}$ dimension across samples, $\hat{\mathbf{y}}_{:,i} \in R^M$.

The loss $L$ is formulated as $L = Y - \hat{\mathbf{Y}}$. The dimension-specific loss $L_{all}$ is computed as $L_{all} = Y/N - \hat{\mathbf{Y}}_{all}$; $L_{all} \in R^{M*N}$. $l_{ij}$ represents the dimension loss for the $i^{th}$ sample in the $j^{th}$ dimension. $l_{:,i}$ indicates all losses for the $i^{th}$ dimension across samples, $l_{:,i} \in R^M$.

The $Relation(i, j) = Cov(l_{:,i}, \hat{\mathbf{y}}_{:,j})$ quantifies the relationship between dimensions $i$ and $j$. A higher value of $Relation(i, j)$ suggests a stronger potential relationship between these dimensions.

|       | 5 segmentation | 10 segmentation | 20 segmentation |
|-------|----------------|-----------------|-----------------|
| Loss  | 0.5415         | 0.1973          | 0.0199          |

Table 1: Single-node CLF experiment result.

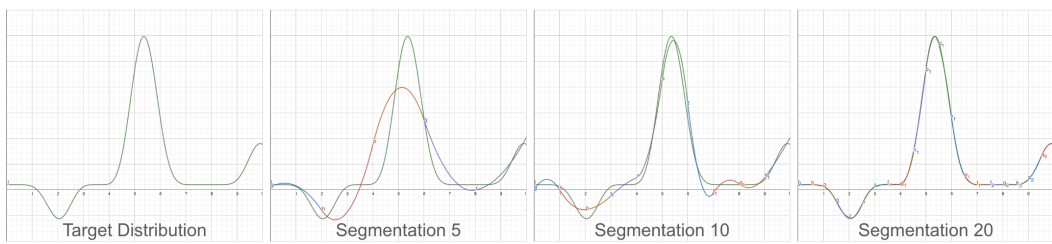

Figure 2: Single-node CLF experiment visualization.

The node relation matrix is calculated as $[i, j, Relation(i, j)]; 0 < i, j < N$. This matrix facilitates the grouping of related nodes based on their interrelationships.

### 2.3.2 MULTI-LAYER STRUCTURE

In the multi-layer CLF, each group of dimensions identified in the previous step is organized into a tree structure. In single-layer CLF, each dimension is represented by a single curve solely dependent on its variables. In multi-layer CLF, the root dimension maintains a single curve influenced only by itself. However, each child dimension possesses multiple curves, specifically one for each segment curve of its parent dimension, and the shape of these child curves depends on both the child variables and its parent segment curve variables. See Fig 1 Right. In terms of structural data, the control list for child dimension in multi-layer CLF is modified to $conList \in R^{N*seg^{layer}*(seg+2)}$, and the equation list is modified to $equList \in R^{N*seg^{layer}*seg*3}$.

## 3 EXPERIMENTS

For the experiments conducted in this section, the CLF model is implemented using Numpy, while the MLP was developed with PyTorch. Firstly, this paper evaluates the CLF model using synthetic mathematical distributions to test its efficiency in fitting the target distribution and in elucidating the relationship between input space and target distribution. Secondly, the performance of CLF is compared to MLP in a taxonomy classification task. Finally, the applicability of CLF to real-world scenarios is assessed using the MNIST dataset.

The CLF model utilizes a quadratic equation list, $equList$, that stores all relationships learned by the module, which can be readily converted into curve images. This paper extensively uses curve images derived directly from the equation list to demonstrate how these curves can be employed to analyze the relationships between input space and target distribution, as well as the interactions among different dimensions. Furthermore, this paper discusses the application of these curves in optimizing the CLF settings and addressing issues related to non-convergence.

### 3.1 SINGLE-NODE CLF: EFFICIENCY AND CAPABILITY

This experiment examines the effect of segmentation numbers on the accuracy of the CLF network. It demonstrates the fitting efficiency and capability across various segmentation levels within single-node CLF configurations. The experiment target distribution is $y = cos^5(0.8x+5)*sin^3(0.4x+3)*(0.2x+7)+0.2; x \in [0, 10]$. The experiment compares the loss value of single-node CLF networks with varying segmentations: 5, 10, and 20. The results are presented in Table 1 and depicted visually in Figure 2.

Table 1 demonstrates that an increase in the segmentation number correlates with a decrease in the loss. Additionally, Figure 2 visually illustrates that a higher segmentation number results more

| Input Length | 5 segmentation | 10 segmentation | 20 segmentation |
|:---:|:---:|:---:|:---:|
| 3-D | 0.5918 | 0.2883 | 0.0351 |
| 4-D | 0.5987 | 0.2932 | 0.0359 |

Table 2: Single-layer CLF experiment.

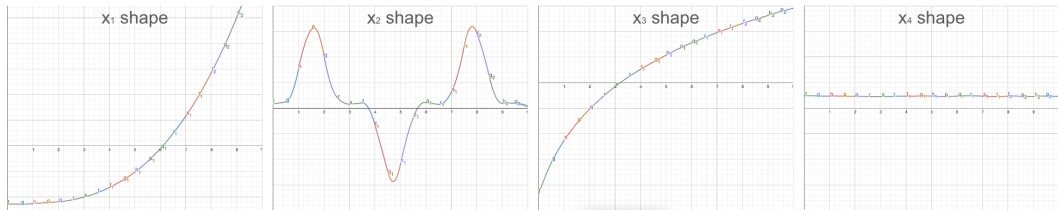

Figure 3: Single-layer CLF experiment visualization.

closely approximating the target distribution curve. This curve can also be interpreted as representing the relationship between the input space and the target distribution. Furthermore, it is possible to directly calculate whether a curve with fewer segments can maintain the same shape. If feasible, this implies that a CLF with fewer parameters could fit the target distribution with the same accuracy.

## 3.2 SINGLE-LAYER CLF: FROM INPUT SPACE TO TARGET DISTRIBUTION

This study assesses the performance of a single-layer CLF model in fitting the distribution $y = f(x_1) + f(x_2)$. It demonstrates how the CLF model captures the relationship between the input space and the target distribution upon completion of training. The target distribution is defined as $y = 0.01x_1^3 + 3sin^5(x_2) + 7log(x_3 + 1) - 6; x_1, x_2, x_3, x_4 \in [0, 10]$. In this setup, $x_4$ acts as a noise dimension. The experiment compares the CLF's loss with different segmentations (5, 10, and 20) and varying input lengths (3 and 4 dimensions). The results are presented in Table 2 and visualized in Figure 3.

Table 2 presents three key findings regarding the performance of the CLF model. First, a single-layer CLF can efficiently fit a target distribution defined by $y = f(x_1) + f(x_2)$. Second, in alignment with prior observations, an increase in the number of segments enhances the model's fitting capability. Lastly, the introduction of a noise dimension impacts the model's fitting accuracy only marginally, by approximately 2-4%.

Figure 3 provides a visualization of the $equList$ for each input dimension, clearly illustrating how CLF effectively discerns the relationship between each dimension and the target distribution. Specifically, $x_1$ corresponds to $0.01x_1^3 + C$, $X_2$, $x_2$ to $3sin^5(x_2) + C$, $x_3$ to $7log(x_3 + 1) + C$, and $x_4$ simply matches $C$. These results demonstrate that the CLF model is capable of isolating and modeling the distinct contributions of various input dimensions to the overall target distribution. A clearly defined curve shape for a dimension suggests its critical role in the model. Conversely, a shape approximating a horizontal line indicates that the dimension has minimal significance.

## 3.3 SINGLE-LAYER VS MULTI-LAYER CLF: INTERACTIONS AMONG DIFFERENT DIMENSION

This experiment compares the performance of single-layer and multi-layer CLF models, examining the effects of various grouping configurations within the multi-layer CLF model. It explores how multi-layer CLF processes and represents the relationships between different input dimensions upon the completion of training. The target distribution used for this experiment is $y = 7sin(x_1) * log(x_2 + 1) + 0.01 * x_3^3 - 5; x_1, x_2, x_3, x_4 \in [0, 10]$. Five CLF models are assessed for their loss: Model 1 is a single-layer CLF; Model 2 is a multi-layer CLF with correct grouping $[[x_1, x_2], [x_3]]$; Model 3 incorporates a noise dimension, grouped as $[[x_1, x_2], [x_3], [x_4]]$; Model 4 is a multi-layer CLF with incorrect grouping $[[x_1, x_2, x_3]]$; and Model 5 is another multi-layer CLF with incorrect grouping $[[x_1, x_3], [x_2]]$. The results are presented in Table 3.

| Segmentation | 5 | 10 | 20 |
|---|---|---|---|
| Single-layer CLF $[x_1, x_2, x_3]$ | 0.9850 | 0.9369 | 0.9389 |
| Multi-layer CLF $[[x_1, x_2], [x_3]]$ | 0.5926 | 0.2684 | 0.1365 |
| Multi-layer CLF $[[x_1, x_2], [x_3], [x_4]]$ | 0.6023 | 0.2786 | 0.1397 |
| Multi-layer CLF $[[x_1, x_2, x_3]]$ | 0.5924 | 0.2658 | 0.1333 |
| Multi-layer CLF $[[x_1, x_3], [x_2]]$ | 0.9602 | 0.9305 | 0.9201 |

Table 3: The experiment results compare the performance of a single-layer CLF with various grouping configurations in multi-layer CLFs.

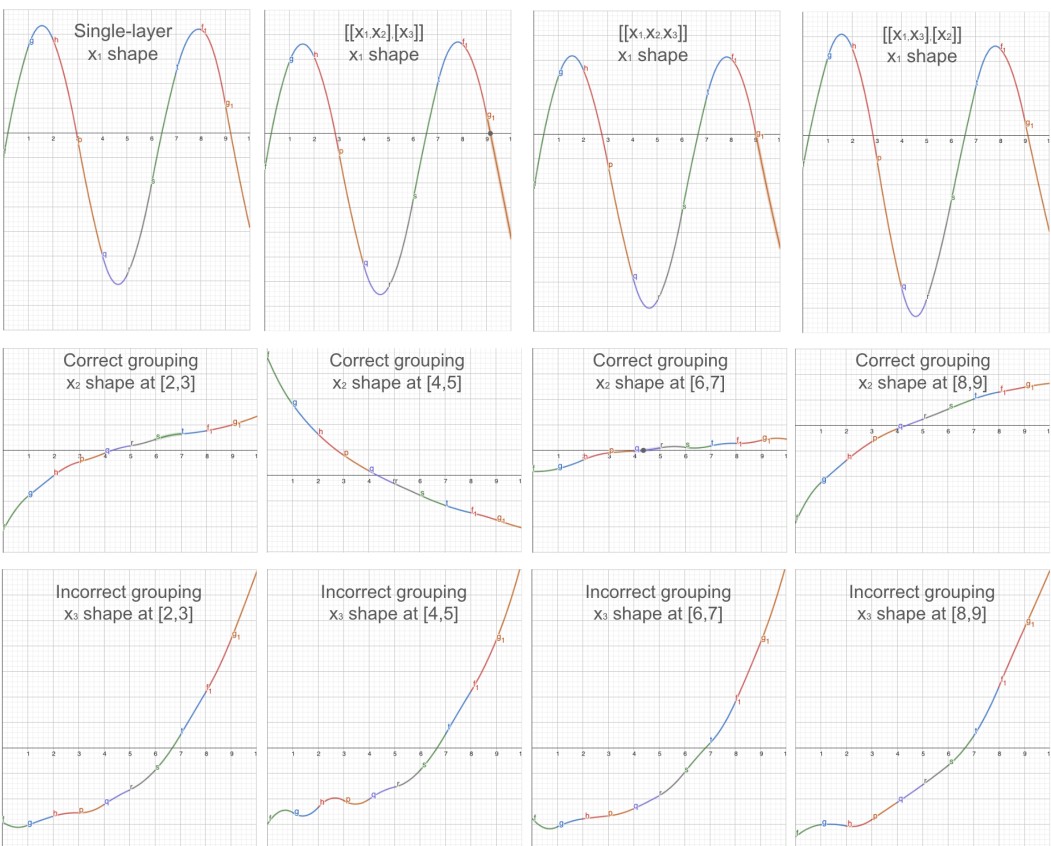

Figure 4: Single-layer VS Multi-layer visualization

Table 3 presents three significant outcomes from the experiment. Firstly, the single-layer CLF is inadequate for efficiently fitting the complex target distribution $y = f(x1, x2)$. This challenge is effectively addressed by employing a multi-layer CLF with correctly grouped input dimensions. Secondly, consistent with findings from the single-layer CLF, increasing the segmentation number in the multi-layer CLF enhances the model's fitting capabilities. Unlike the single-layer CLF, the additional noise dimensions in the multi-layer configuration slightly improve the fitting accuracy. Lastly, while grouping unrelated dimensions does not significantly impact the fitting ability, separating related dimensions into different groups markedly reduces the model's effectiveness in fitting the target distribution.

In the multi-layer CLF structure, the root dimension features a single curve, whereas the child dimensions exhibit multiple curves. Figure 4 illustrates these relationships through a series of comparisons. The first row compares the shape of the root dimension, $x_1$, across single-layer CLF, multi-layer CLF with correct grouping, and multi-layer CLF with incorrect grouping. The second

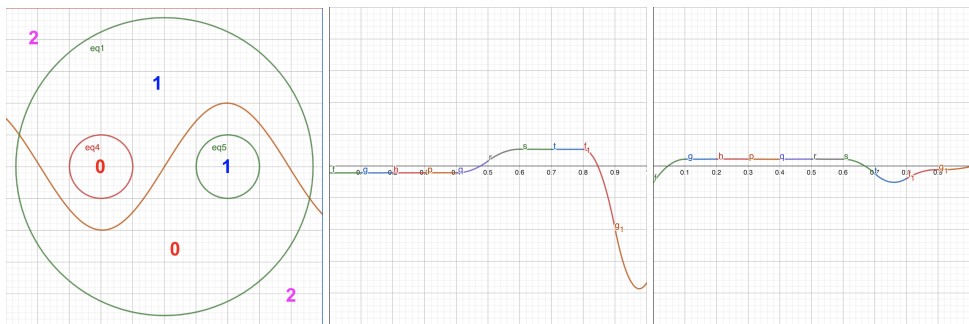

Figure 5: Left: Taxonomy dataset distribution. Middle: 2-layer 10-segment CLF for category 0 root dimension (x-coordinate) curve. Right: 2-layer 10-segment 0-category CLF child dimension (y-coordinate) curve adjusting root $[0.2, 0.3]$ segment curve.

| Segmentation | 3 | 5 | 10 |
|---|---|---|---|
| CLF $[[x_1, x_2]]$ | $86.77 \pm 0.21\%$ | $94.08 \pm 0.13\%$ | $96.15 \pm 0.07\%$ |
| Configurations | 2-6-4-3 | 2-8-6-3 | 2-8-16-6-3 |
| MLP | $83.81 \pm 2.45\%$ | $89.91 \pm 1.46\%$ | $92.91 \pm 5.78\%$ |

Table 4: Taxonomy task experiment result.

row examines the shape of a child dimension, $x_2$, which is related to the root dimension, while the third row focuses on $x_3$, a child dimension not related to the root dimension.

The analysis reveals several key observations. In the first row, the curves corresponding to the root dimension $x_1$ are highly similar across different CLF configurations, indicating a consistent contribution regardless of the model settings. In the second row, where $x_2$ is directly related to the root dimension, the curves exhibit distinct shapes influenced by the root dimension's behavior. Specifically, the original feature $log(x_2 + 1)$ is modified by a coefficient derived from the root dimension's value. For instance, during segments $[2, 3]$, the root value is positive, keeping the $x_2$ curve as $log(x_2 + 1)$. In segments $[4, 5]$, where the root value is negative, the $x_2$ curve inverts. In segments $[6, 7]$, with the root value around zero, the $x_2$ curve appears squeezed, and in segments $[8, 9]$, similar to segments $[2, 3]$, the curve retains its original shape. These variations demonstrate that a multi-layer CLF with correctly grouped dimensions can significantly enhance the model's fitting ability. Conversely, in the third row, the unrelated child dimension $x_3$ shows similar curve shapes across different root values, reflecting its independence from the root dimension. The $x_3$ curves consistently represent its inherent feature, $0.01 * x_3^3$, unaffected by the root dimension's fluctuations. This consistency allows the determination of whether there is a relationship between child and root dimensions by comparing the shapes of the child dimension curves.

### 3.4 TAXONOMY CLASSIFICATION: CLF VS MLP

This study conducts a comparative analysis between CLF and MLP on a taxonomy classification task involving three categories in two dimensions. The target distribution for this experiment is illustrated in Figure 5 left, with the variable range for $x_1$ and $x_2$ set between 0 and 1. The experiment assesses different configurations of 2-layer CLF with segmentation numbers of 3, 5, and 10, corresponding to 90, 168, and 468 parameters, respectively. For a comparison in terms of model complexity, the MLP configurations are adjusted to 2-6-4-3 with 96 parameters, 2-8-6-3 with 164 parameters, and 2-8-16-6-3 with 516 parameters. All MLP uses ReLU activation function. After completing the training process, an additional 10 iterations are conducted. From these iterations, the average value and the maximum deviation from the average are calculated. The outcomes of these configurations are detailed in Table 4.

Although both the CLF and MLP models in this experiment have a comparable number of parameters, the author does not consider this a fair comparison. In MLPs, the forward pass requires the

|  | CLF 1-L | CLF+ 1-L | CLF 2-L | CLF+ 2-L | MLP 784-10 | MLP 784-480-10 |
|---|---|---|---|---|---|---|
| Training | 96.93% | 95.18% | 99.97% | 98.61% | 92.90% | 99.15% |
| Test | 90.73% | 92.85% | 94.97% | 95.67% | 92.37% | 97.92% |

Table 5: MNIST experiment result.

involvement of all parameters in the computation, and the optimization process updates all parameters. The Method section details the operation of CLF, where the forward pass only necessitates one quadratic equation per input dimension, and optimization updates merely three control points for each input dimension. This efficiency arises because CLF opts for a trade-off of larger memory usage in exchange for reduced computational demand. Consequently, equating the two models based solely on the number of parameters places CLF at a disadvantage. When parameter counts are equal, CLF operates significantly faster than MLP, particularly in larger models.

Table 4 presents multiple findings from the experiment comparing CLF and MLP. Firstly, CLF demonstrates greater stability than MLP. During the experiment, multiple MLP models were retrained due to non-convergence. Even among those that did converge, it was challenging to ascertain whether they had achieved optimal performance. In contrast, each CLF was trained only once, and upon completion, yielded highly consistent results, with deviations from the average value ranging only from 0.07% to 0.21%. In comparison, MLP results varied from the average by 1.46% to 5.78%.

Secondly, despite having a similar number of parameters, MLPs consistently showed lower accuracy than CLFs. Thirdly, CLF not only demonstrated superior accuracy but also operated significantly faster than MLP in both the forward pass and optimization phases.

Lastly, CLF's ability to visually represent the relationships it learns is notably advantageous. Figure 5 illustrates this with two images: the middle image depicts the root dimension (x-coordinate) shape of a 2-layer, 10-segmentation CLF model for category 0, while the right image shows the corresponding child dimension (y-coordinate) shape. These images demonstrate how the root dimension influences the categorization, indicating that category 0 is likely when $x$ is within the range [0.48, 0.85]. Despite the root dimension suggesting the absence of category 0 for $x$ values in the range [0.2, 0.3], adjustments in the child dimension for $y$ values in the range [0.1, 0.6] also result in category 0. This capacity to depict learned relationships is something that MLP lacks, highlighting a distinct advantage of CLF in providing interpretable results.

### 3.5 MNIST: CLF IN REAL-WORLD TASK

This experiment evaluates the effectiveness of CLF models in a real-world classification task using the MNIST dataset. It compares the performance of MLP, standard CLF, and CLF+. Specifically, the experiment involves training 1-layer CLF and CLF+ models, each with 3 segmentations across 784 input dimensions. Upon completion of training, the 1-layer CLF model is used to identify and eliminate non-essential input dimensions based on their importance. Subsequently, 2-layer CLF and CLF+ models are trained using 3 segmentations but with reduced input dimensions, fewer than 400. In contrast, the MLP models are configured with two different architectures: one with a single layer of 10 neurons (784-10) and another with two layers containing 480 and 10 neurons respectively (784-480-10). The results of these comparisons are presented in Table 5.

The analysis of Table 5 yields several insights. Firstly, CLF demonstrates higher accuracy on the training dataset but lower accuracy on the test dataset than MLP. This suggests that while CLF can fit the training data more precisely, it lacks the generalizability of MLP. Secondly, there is a noticeable increase in overfitting issues as the layer number of CLF is increased. Lastly, the CLF+ model mitigates these overfitting problems, indicating an improvement in model robustness. Due to space limitations, further discussion of generalizability issues is provided in the Appendix.

### 3.6 EXPERIMENTS SUMMARY

The CLF model, by utilizing Bezier Curves, creates a network structure tailored to fit the target distribution and eliminates the need for activation functions when addressing nonlinear distributions.

The control points of the Bezier Curve are further converted into quadratic lists, which store and display the relationships learned by the model. This approach renders the CLF model fully interpretable and facilitates the clear presentation of the relationships or knowledge it has acquired.

Firstly, the CLF model's ability to present learned relationships is demonstrated through experiments. The single-layer CLF experiment visually confirms that the CLF can efficiently identify the relationship between the input space and the target distribution (Fig 3). The multi-layer CLF experiment visually confirms that the CLF can effectively discern the interactions between the root and child dimensions (Fig 4).

Moreover, the CLF interpretability provides valuable guidance during and after training. In the single-node CLF experiment, the model's capability is assessed (Fig 2), aiding in determining the minimal CLF structure necessary to represent a relationship. In the multi-layer CLF experiment, issues such as incorrect dimension grouping leading to convergence problems (Table 3) or inefficient use of parameters ((Fig 4)) are identified. These issues are detected either through non-convergence of the model or by comparing different child dimensions' curves, highlighting areas where improvements can be made.

## 4    CONCLUSION

The CLF model offers two primary advantages. Firstly, it is fully transparent and explainable, efficiently illustrating the relationship between input space and target distribution, the contributions from different dimensions, and the interactions between these dimensions. Secondly, CLF provides a clear guideline on how to initialize the model. Upon completion of training, the model allows for the evaluation of the necessity of each segment curve. If the dimension curve can be represented with fewer segments, then the number of segments should be reduced. Similarly, if child curves present similar shapes, they should be removed from their parent structure. Despite its effectiveness in fitting the target distribution accurately, CLF still encounters several challenges that need addressing, including issues related to generalizability, grouping accuracy, and potential overfitting.

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

# A APPENDIX

## A.1 METHOD APPENDIX

### A.1.1 FORMULAS DERIVATION: GET $\hat{\mathbf{y}}$

The Bezier Curve equation is given by $y = x_1 + (x_2 - x_1)t = (1-t)x_1 + tx_2, t \in [0, 1]$. Previous example present that each segment curve is controlled by three control points, denoted as $P_1$, $P_2$, and $P_3$. The midpoints between these control points are calculated as $P_{12} = \frac{P_1 + P_2}{2}$ and $P_{23} = \frac{P_2 + P_3}{2}$. To compute the parameter $t$, which determines the specific point on the Bezier Curve, the formula is: $t = \frac{x - P_{12x}}{P_{23x} - P_{12x}}$. Given that all control points have fixed x-positions, the span $s$ can be expressed as: $s = P_{23x} - P_{12x} = \frac{input\ range}{segmentation\ number}$. Thus, the parameter $t$ simplifies to $t = \frac{x - P_{12x}}{s}$. Utilize $Bez()$ implement Bezier Curve function:

$$Bez(P_{12}, P_2) = (1-t)P_{12} + tP_2; Bez(P_2, P_{23}) = (1-t)P_2 + tP_{23}$$

$$\hat{\mathbf{y}} = Bez(P_{12}, P_2, P_{23}) = (1-t)Bez(P_{12}, P_2) + tBez(P_2, P_{23})$$

$$= (1-t)[(1-t)P_{12} + tP_2] + t[(1-t)P_2 + tP_{23}]$$

$$= (1-t)^2 P_{12} + 2t(1-t)P_2 + t^2 P_{23}$$

$$= (P_{12} - 2P_2 + P_{23})t^2 + (-2P_{12} + 2P_2)t + P_{12}$$

$$= (\frac{P_1 + P_2}{2} - 2P_2 + \frac{P_2 + P_3}{2})t^2 + (-2\frac{P_1 + P_2}{2} + 2P_2)t + \frac{P_1 + P_2}{2}$$

$$= (\frac{P_1}{2} - P_2 + \frac{P_3}{2})t^2 + (-P_1 + P_2)t + \frac{P_1 + P_2}{2}$$

### A.1.2    CLF+

The CLF model is proficient in modeling target distributions, a feature that frequently leads to pronounced overfitting issues. Various methods have been proposed to mitigate this problem, including early stopping [Lodwich et al. (2009)] and momentum [Jelassi & Li (2022)]. Inspired by federated learning [Zhang et al. (2023)], which involves comparing local network parameters before generating a master optimized value for updating the server network, CLF+ divides the training dataset into several local datasets and trains corresponding local networks. Both CLF and CLF+ utilize identical network architectures; however, they are differentiated by their respective training methodologies.

Before Training: A comparison group number, denoted as $cpr$, is established before training. The training dataset $X$ is then divided into $cpr$ groups, each serving as a local dataset containing a similar amount of samples across categories. Concurrently, $cpr + 1$ instances of the CLF models are initialized with identical initializations. This ensures that all CLF models begin with the same configuration, specifically having the same $conList$ matrix.

Training: Initially, one of the CLF models is designated as the master network, while the remaining $cpr$ CLFs are classified as local networks. Each local network is paired with a corresponding local dataset for training purposes. All local networks undergo one iteration of training, after which each local network's $conList$ matrix will have diverged from that of the master network. The subsequent step computes an optimization value matrix. This is achieved by subtracting the master network's $conList$ from each local network's $conList$, resulting in $cpr$ optimization value matrices. For each optimized value in the $cpr$ matrices, if all $cpr$ optimization values are positive, the minimum value is selected; if all are negative, the maximum value is chosen; if there is a mix of positive and negative values, zero is selected. These selected values are then used to construct the final optimization value matrix. Lastly, this final optimization value matrix is added to the master network's $conList$, forming the new master $conList$. This updated $conList$ is then broadcast to all local networks. Subsequently, the next iteration of training commences.

This training strategy is designed to effectively regulate the overfitting issues. A higher $cpr$ results in low overfitting, whereas a lower $cpr$ count leads to high overfitting. Further elaboration on the CLF+ training strategy is provided in the "Discussion Federated Learning Solutions to Overfitting" section.

### A.2    DISCUSSION

#### A.2.1    INPUT RANGE LIMITATION

MLP utilize linear equations, allowing them to cover the entire numerical range of inputs easily. Conversely, CLF models employ Bezier Curves, which inherently have defined start and end points, limiting their ability to cover the entire numerical range. To address this limitation, three potential solutions are proposed:

Input Space Condensation: Although MLP models typically cover a broad input range, in practice, the input space is often condensed into a smaller range [Ioffe & Szegedy (2015); Patro & Sahu (2015)] to enhance computational efficiency and prevent gradient explosions. This method of condensing the input space can also be applied to CLF models. By scaling and translating input data into a manageable range, the efficiency of CLF can be improved without altering its underlying architecture.

Dynamic Range and Segmentation Adjustment: CLF models require a predefined input range and segmentation number. If inputs fall outside the established range, the model can extend both the input range and the number of segments without altering the existing configuration. For instance, a CLF model with an initial range of [0, 10] and five segments can be expanded to cover [0, 12] with six segments. This adjustment ensures that the original input range remains unchanged, and the newly added segment curve shape from [10, 12] can be integrated using existing network parameters from the [0, 10] range.

Transformation Methods: Techniques such as hyperbolic transformations [Rader & Steinhardt (1986)] can shift and scale the entire numerical range to fit within a specific interval. Applying such transformations to CLF models can enable them to handle inputs across the entire numerical spectrum, thereby enhancing their applicability and flexibility.

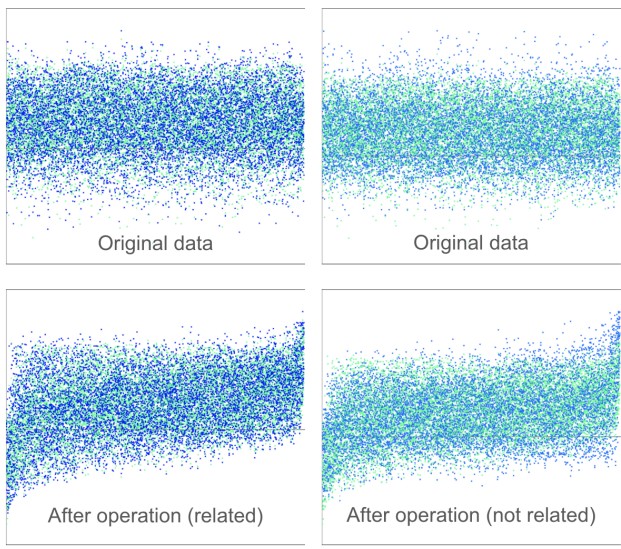

Figure 6: After certain operations, related dimension present the similar distribution.

These strategies collectively enable CLF models to overcome their inherent limitations regarding input range, making them more versatile for various applications.

### A.2.2  GROUPING

This grouping theory is inspired from the gradient boosting theory. It posits that in a function such as $y = f(x_1, x_2)$, the variables $x_1$ and $x_2$ must be capable of compensating for each other's negative loss. To derive Equation $Relation(i, j)$, the method initially calculates the negative loss and the estimated $\hat{\mathbf{y}}$ distribution for each dimension. Subsequently, it determines the relationship score between dimensions using the covariance method, as visualized in Figure 6.

It is important to note that a higher relationship score indicates a greater likelihood of a relationship between two dimensions, but it does not conclusively prove a connection. It is feasible for different groups to exhibit similar distributions, which may lead to the erroneous grouping of unrelated dimensions. For instance, in the target distribution $y = x_1x_2 + x_3x_4$, $x_1$ and $x_2$ form one group, and $x_3$ and $x_4$ form another; however, all four dimensions display identical distributions. To address this potential misclassification, Figure **??** proposes a method to confirm the genuine relationships within a group.

### A.2.3  OPTIMIZE NETWORK ONLY WHEN TASK FAIL

In the CLF model, the optimization function is activated exclusively in instances of task failure. This operational strategy is underpinned by two principal reasons. Firstly, relying on a single numerical target fails to provide absolute right directional guidance for model training. As indicated in the distillation study [Hinton (2015)], the model utilizes probabilities from the teacher model as soft targets and categorical numbers as hard targets. The findings from this study demonstrate that soft targets offer more precise directional guidance and are less prone to overfitting than hard targets. While a single categorical number can provide a general training direction, it does not furnish an absolute right correct path. Therefore, the optimization function in CLF is triggered only when a task fails. Secondly, the CLF model forward function demands minimal computational resources, whereas the toQuadraticList function is extremely resource-intensive. Given this, CLF predominantly employs the forward function, optimizing training efficiency and minimizing training duration by reserving the toQuadraticList function for occasions when the task fails.

