# OpenReview forum: "CLF: Curve Line Fitting Neural Network Based On Bezier Curve"
_ICLR.cc/2025/Conference — Submitted to ICLR 2025_

### Official Review · Reviewer_j96p · 2024-10-26

**Soundness:** 3
**Presentation:** 2
**Contribution:** 2
**Rating:** 3
**Confidence:** 3

**Summary:**

This work aims to learn and construct a nonlinear mapping using Bézier curves, departing from the traditional paradigm of neural network activation functions. The study is both novel and interesting. I encourage further exploration in this area; however, the overall research appears to lack rigor, and several issues remain unaddressed.

**Strengths:**

The questions are both profound and challenging.

**Weaknesses:**

1. Several critical issues remain unaddressed.
2. The experiment was insufficient and lacked rigor.
3. Please refer to the questions.

**Questions:**

1.	In line 98, it states that “the input space X is evenly divided into five segments, governed by control points labeled A through G.” However, I cannot find the corresponding five segments in Figure 1. Please clearly label or highlight the five segments in Figure 1.
2.	In line 116, how is $P_{12x}$ defined?
3.	Is this approach susceptible to overfitting issues? How does CLF address overfitting in its implementation? Authors could add a specific section discussing overfitting in CLF, including any experiments or comparisons with traditional neural networks in terms of overfitting behavior.
4.	Neural networks can effectively manage the decoupling relationship between multiple features, eliminating the need for complex feature engineering. How are multiple feature inputs handled in this method? From the text's description, it seems to involve simple summation at the output. CLF does not appear to be well-equipped to handle multi-feature tasks efficiently. It is suggested that the authors provide a detailed explanation or illustration of how CLF handles multiple features, possibly with a comparison to how traditional neural networks manage this.
5.	In section 2.3, a multi-layer CLF is mentioned. However, based on the description, this approach does not seem to be end-to-end. Such a method may not be effective in practical applications, particularly for constructing deeper networks.
6.	Can CLF be extended to classification tasks? How is the loss function defined for classification?
7.	I believe the operational complexity of this approach could be significant when dealing with high-dimensional inputs and outputs. For instance, in the MINST experiment mentioned in the paper, how does the running complexity of CLF compare to that of multilayer perceptron models? We suggest that the authors include a detailed complexity analysis section, comparing the time and space complexity of CLF to traditional neural networks for various input dimensions, particularly focusing on the MNIST experiment mentioned in the paper.

---

### Official Review · Reviewer_ZwZu · 2024-11-02

**Soundness:** 1
**Presentation:** 1
**Contribution:** 1
**Rating:** 1
**Confidence:** 5

**Summary:**

The author(s) introduced a novel hierarchical method to fit the functions. In their main contribution, the authors emphasizes that their model  doesn't rely on the nonlinearity from the activation function as the conventional multilayer neural networks. Instead, they developed a sequential fitting algorithm with Bezier Curve. In each step, they tune the control points to minimizes the regression error. The author compares the performance of CLF and MLP in synthetic data and real world data.

**Strengths:**

The paper wraps up a sound theory by developing a sequential fitting algorithm with Bezier Curve. It details the construction of the Bezier Curve and make a systematic comparison between CLF and MLP in synthetic data and real world data.

**Weaknesses:**

The paper needs some improvements in writing, including
1. The algorithm in section 2.1.6 should format in an algorithm setup. It also should include a short sentences for each of the abbreviations, like LR for learning rate and etc.
2. Nonlinear distribution. The paper first mentions the terminology 'nonlinear distribution' in line 087 without correct reference or definition  for the term. Usually, distribution refers to the probability distribution, while in this paper it seems like 'nonlinear function class' would be a better description.
3. The abuse of the notations. In line 120, the author(s) uses $P_1'$ to denote the derivative with respect to $P_1$, which is not rigorous. The variable should be a subscript conventionally.
The paper lacks of evidence why the sequential CLF is interested,
1. CLF doesn't show theoretical soundness compared with MLP. The modern approximation theory for MLP can (partially) explain why and how the neural networks works, including the universal approximation theory, the sample complexity and etc.. The lack of fully developed theory is not a solid reason for developing a new method. Also, in a lot statistical problems, the performance of traditional methods are comparable with that of the neural networks.
2. The results demonstrated in table 5 only indicates the comparable performance of CLF (or even worse). It lacks of the evidence when and why CLF is superior than MLP.

**Questions:**

I might miss something though I have several concerns about the paper,
1. If by increasing the number of control points, CLF can achieve better regression results. Then what's the motivation to develop multiple layer CLF? Also, In traditional fitting problem, dividing the domain into subdomain and solve the same fitting problem on the subdomain correspondingly will yield a better regression, is it necessary to discuss the reason why choose to use hierarchical structure instead of the windowed fitting?
2. Does CLF also have universal approximation theory? What's the convergence rate with respect to the number of control points.
3. Can CLF generalize the results to more complicated function class, beyond the smooth functions? like the function with regularities.
4. The proposed CLF seems like to address the interpretability difficulty of MLP, but what exactly the difficulty does CLF try to solve? like sample complexity or the convergence rate, or the meaning of the parameters?

---

### Official Review · Reviewer_ELgY · 2024-11-04

**Soundness:** 2
**Presentation:** 2
**Contribution:** 2
**Rating:** 3
**Confidence:** 3

**Summary:**

This paper proposes a Curve Line Fitting (CLF) network, which uses Bezier curve fitting to fit nonlinear relations compared to linear regression with activation functions in MLP. The authors claim that this CLF model is more interpretable than MLP.

**Strengths:**

- propose to replace traditional linear regression in MLP with Bezier curve regression, which can remove the using of activation functions and be more interpretable.
- Some initial experiments are conducted to show the efficacy of the proposed method.

**Weaknesses:**

- The experiment is only performed on small synthetic datasets and MNIST. The test accuracy on MNIST is very low compared with current models that can achiever almost perfect accuracy.
- I don't see how the proposed CLF is more interpretable than MLP. Instead of learning linear combinations of input in MLP, CLF is learning quadratic functions over each dimension and then sum over all dimensions. The nonlinear function of each dimension is itself not interpretable.
- See some other question below

**Questions:**

- The Bezier Curve is not introduced in the paper. I think the authors use the Quadratic Bézier curves? It would be better to have some introduction of Bezier Curve.
- A single node CLF can only calculate quadratic functions of each dimension of the data and there is no interaction between different dimensions. Does it have enough expressive power to represent complex functions?
- Line 211 $\hat{Y}$ is not defined. Line 212, $Y \in R^M$ and $\hat{Y}_{all} \in R^{M \times N}$. How do you calculate $Y - \hat{Y}\_{all}$?
- What is $Relation(i, j)$ and $Cov$ operation?

---

### Official Review · Reviewer_cm5c · 2024-11-08

**Soundness:** 2
**Presentation:** 3
**Contribution:** 1
**Rating:** 3
**Confidence:** 4

**Summary:**

This paper proposes to use Bezier curve regression to replace the linear regression in a multi-layer perceptron (MLP), and also remove all activations, in order to realise (1) more efficiently fitting target distributions, and (2) “insights into the relationships between input dimensions and target distributions, as well as the interdependencies across different dimensions” I quoted.

**Strengths:**

Pros:

- The paper presents detailed introduction and explanation of the proposed method, including estimator of the label, optimisation, forward function, initialisation, training process for cases of single node, single layer, and multiple layers. The paper is compressive, well-structed, and easy to follow.

- The paper present comprehensive experiments for cases of single node, single layer, and multiple layers, in terms of efficiency, capability, mapping from input to the distribution estimated, and interaction between different dimensions.

- I find the paper makes conceptual contributions of proposing a new family of neural networks that could have advantages.

**Weaknesses:**

Cons:

- “Extraordinary claims require extraordinary evidence.” Proposing a new family of neural networks needs strong evidence in their advantages with existing methods, in either theoretical or empirical aspect, or more ideally, in both. Unfortunately, this paper fails to persuade me they are better. The experiments are too simple – only comparing the method with MLP using MNIST. No theoretical analysis is provided. I would suggest including experiments on at least CIFAT and ImageNet, comparing with CNN, ResNet, and transformer. For theoretical studies, I would encourage analyse the generalisability, approximation, and stability of the proposed method. For example, does the hypothesis space of CLF has smaller approximation error than existing methods? Does the model leaned by CLF has a smaller generalisation bound?

- The paper claims that the CLF is “fully interpretable”. I challenge this. First, I guess the authors meant to say “explainable” rather than “interpretable”. The two terms may have similar lay-man meanings, but in machine learning, “interpretable” is usual for interpreting model mechanisms (such as the theory of deep learning), but “explainable” is for explaining different data features’ contributions. Second, this paper attributes the explainable issues of MLP to nonlinear activations, and thus CLF is fully explainable because it removes activations. This is groundless. To render a solid discussion, please (1) define what is explainable, (2) explain why activation make it unexplainable, and (3) show how to explain the results of CLF.

**Questions:**

Please address the Weaknesses.

---

### Meta-Review · Area_Chair_JM3B · 2024-12-20

**Metareview:**

This paper proposes using Bezier curve regression to create a multilayer network and introduces a training strategy to learn its parameters from data. A key benefit of this approach is that it eliminates the need for activation functions, as Bezier curves can directly model nonlinearity. The authors claim this leads to a more interpretable model, though some reviewers challenged this assertion. Overall, the proposed idea is radically different from conventional methods of constructing and training neural networks, and thus has the potential for impactful contributions and may open new research directions in the field. However, the current draft is not mature enough for publication and requires further development.

While reviewers appreciated the novelty of the proposed idea, they provided suggestions to strengthen the paper's claims and make them more convincing. All reviewers expressed concerns about the limited experiments demonstrating the usefulness of the approach, as the presented results are based on synthetic data or limited to MNIST. Reviewers cm5c and ELgY questioned the interpretability claims made by the authors. The authors did not respond to these comments, leaving the concerns unaddressed. Given the paper's current state and the feedback received, it cannot be accepted in its present form. However, considering the originality of the idea, it could eventually lead to a significant contribution. I encourage the authors to resubmit their work after revising it based on the feedback received during this review cycle.

**Additional Comments On Reviewer Discussion:**

While reviewers appreciated the novelty of the proposed idea, they provided suggestions to strengthen the paper's claims and make them more convincing. All reviewers expressed concerns about the limited experiments demonstrating the usefulness of the approach, as the presented results are based on synthetic data or limited to MNIST. Reviewers cm5c and ELgY questioned the interpretability claims made by the authors. The authors did not respond to these comments, leaving the concerns unaddressed. Given the paper's current state and the feedback received, it cannot be accepted in its present form. However, considering the originality of the idea, it could eventually lead to a significant contribution. I encourage the authors to resubmit their work after revising it based on the feedback received during this review cycle.

---

### Decision · Program_Chairs · 2025-01-22

Reject